# Revealing the complexity of ionic liquid–protein interactions through a multi-technique investigation

Liem Bui-Le [1], Coby J. Clarke [1], Andreas Bröhl[1], Alex P. S. Brogan [2], James A. J. Arpino[1], Karen M. Polizzi [1] & Jason P. Hallett [1✉]

Ionic liquids offer exciting possibilities for biocatalysis as solvent properties provide rare opportunities for customizable, energy-efficient bioprocessing. Unfortunately, proteins and enzymes are generally unstable in ionic liquids and several attempts have been made to explain why; however, a comprehensive understanding of the ionic liquid–protein interactions remains elusive. Here, we present an analytical framework (circular dichroism (CD), fluorescence, ultraviolet-visible (UV/Vis) and nuclear magnetic resonance (NMR) spectroscopies, and small-angle X-ray scattering (SAXS)) to probe the interactions, structure, and stability of a model protein (green fluorescent protein (GFP)) in a range (acetate, chloride, triflate) of pyrrolidinium and imidazolium salts. We demonstrate that measuring protein stability requires a similar holistic analytical framework, as opposed to single-technique assessments that provide misleading conclusions. We reveal information on site-specific ionic liquid–protein interactions, revealing that triflate (the least interacting anion) induces a contraction in the protein size that reduces the barrier to unfolding. Robust frameworks such as this are critical to advancing non-aqueous biocatalysis and avoiding pitfalls associated with single-technique investigations.

[1] Department of Chemical Engineering, Imperial College London, London SW7 2AZ, UK. [2] Department of Chemistry, King's College London, Britannia House, London SE1 1DB, UK. ✉email: j.hallett@imperial.ac.uk

Due to their highly tunable nature, ionic liquids are becoming increasingly popular as potential solvents for a variety of applications in biocatalysis, extraction, and electrochemistry[1,2]. The attractiveness of using these solvents instead of aqueous buffered conditions, particularly in industrial biocatalysis, stems from their ability to solvate a broader range of organic substrates, and that their negligible vapor pressure significantly reduces the energy requirements for product purification[3,4]. However, biocatalysts are typically unstable and poorly soluble in solvent systems other than aqueous buffered conditions[5]. Strategies to remove water entirely, such as solvent-free biofluids of in neat (dry) ionic liquids[6,7], can provide a platform for high enzyme solubility (>50% by mass) with a significant increase in thermal stability. Nevertheless, the activity of some enzymes is improved in ionic liquid–water mixtures, most notably these include lipase-catalysed esterification or hydrolysis[8–10].

Despite significant advances in biotechnology, the interactions between proteins and ionic liquids are still not fully understood. The structural diversity of both ionic liquids and proteins mean that a wide range of solvent-protein interactions is possible, making it challenging to draw accurate and general conclusions. The Hofmeister series is often invoked to explain the interactions of ionic liquids with proteins[11,12], which tend to be protein-dependent[13] and often contradictory[4,14]. This contradictory nature of how ionic liquids interact with proteins is manifested acutely in the activity of enzymes, where different behaviours may be observed depending on whether the enzyme is free, cross-linked, or immobilized on a solid support[15]. Furthermore, the properties of ionic liquid–water mixtures may influence protein activity. For example, micelle formation has been shown to induce superactivity of α-chymotrypsin in the presence of long alkyl chain ionic liquids[16].

Protein stability in ionic liquids is a function of ionic liquid composition (including presence or lack of functional groups (e.g., hydrogen bonding hydroxyls), protein surface composition and charge, pH (in aqueous mixtures), and specific cation/anion effects (e.g., hydrogen bond disruption)[17–20]. Additionally, protein stability can be rationalized via site-specific interactions between amphiphilic structures (e.g., electrostatic vs hydrophobic character) and overall solvent properties. All of these parameters must be considered to fully assess the impact of an individual ionic liquid on the structure and stability of any protein. This creates a complex problem that a single thread of enquiry (such as fluorescence spectroscopy) may lack the nuance to assess fully. Instead, this requires a multifaceted investigation involving the use of complementary techniques, such as multiple spectroscopies and scattering, to truly determine the impact of ionic liquids on protein structures.

Here, taking full advantage of the highly characterized — in terms of secondary and tertiary structure — green fluorescent protein (GFP), we provide an analytical framework for a comprehensive investigation into the interactions between ionic liquids and proteins (Fig. 1a). Using ultraviolet–visible (UV/Vis), fluorescence, circular dichroism (CD), and nuclear magnetic resonance (NMR) spectroscopies (for detailed structural analysis) alongside small-angle X-ray scattering and thermal denaturation studies, we have thoroughly investigated the specific and non-specific interactions between GFP and a range of 1-butyl-1-methylpyrrolidinium ([bmpyrr]) and 1-butyl-3-methylimidazolium ([bmim]) salts (Fig. 1b).

This has enabled us to address the seemingly contradictory information that many previous studies have given[21–23]. Our multi-technique approach demonstrates the need to consider the mutual effects of solvent-on-protein and protein-on-solvent, avoiding the overinterpretation of one line of enquiry and

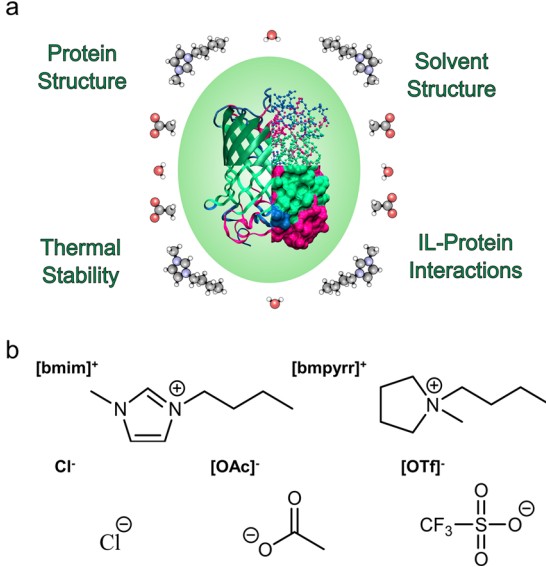

**Fig. 1 Ionic liquid–protein interactions probed and ionic liquids used in this study. a** Properties and interactions probed by the multi-technique approach for a holistic understanding of proteins in ionic liquid solutions. **b** Structures of the cations and anions, with their abbreviations above the structure, studied in this work.

emphasising that structure does not necessarily equate to stability and vice versa. We hope that future studies along these lines will facilitate a greater understanding of the rich, complex interplay between ionic liquids and proteins, establish a basis for swift improvement of process design, and enable strategies to better assess optimal conditions for the full realization of ionic liquids as viable reaction media for industrial biocatalysis.

## Results and discussion
**Structure of GFP in aqueous ionic liquids solutions**. The first stage of assessing the interaction between ionic liquids and proteins is to determine the impact the ionic liquid has on the structure of the protein. The anions selected for this study ([OAc]$^-$, Cl$^-$, and [OTf]$^-$) were due to their water-soluble nature and the broad range of coordination ability as determined from Kamlet-Taft β parameters being 1.18, 0.87 and 0.49 for [OAc]$^-$, Cl$^-$, and [OTf]$^-$ respectively[24,25]. Here, we used CD and UV/Vis spectroscopies to investigate the influence of ionic liquid solutions on the secondary and tertiary structures of GFP, respectively. For CD spectroscopy, we were limited to pyrrolidinium salts, as imidazole has significant absorbance in the far-UV region. CD spectra for GFP in water and aqueous solutions of [bmpyrr][OAc], [bmpyrr]Cl, and [bmpyrr][OTf] (Fig. 2a) all showed a negative band at 215 nm, characteristic of predominantly β-sheet secondary structure. Furthermore, the intensity of the CD spectrum signal plots for the aqueous and ionic liquid solutions were all comparable, highly suggestive that the pyrrolidinium ionic liquids did not significantly impact the secondary structure. Similarly, UV/Vis spectroscopy (Fig. 2b and Supplementary Fig. 2a) showed that the $\lambda_{max}$ associated with the GFP chromophore remained unchanged at 487 nm across the aqueous and the ionic liquid solutions. Given the dependence of the absorbance intensity at this $\lambda_{max}$ on the tertiary structure of GFP, these results suggested that there was a negligible impact of the ionic liquids on protein tertiary structure.

Fluorescence intensity of the GFP fluorophore has a greater sensitivity to perturbations in the protein structure; as such, fluorescence spectroscopy was used to complement the UV/Vis

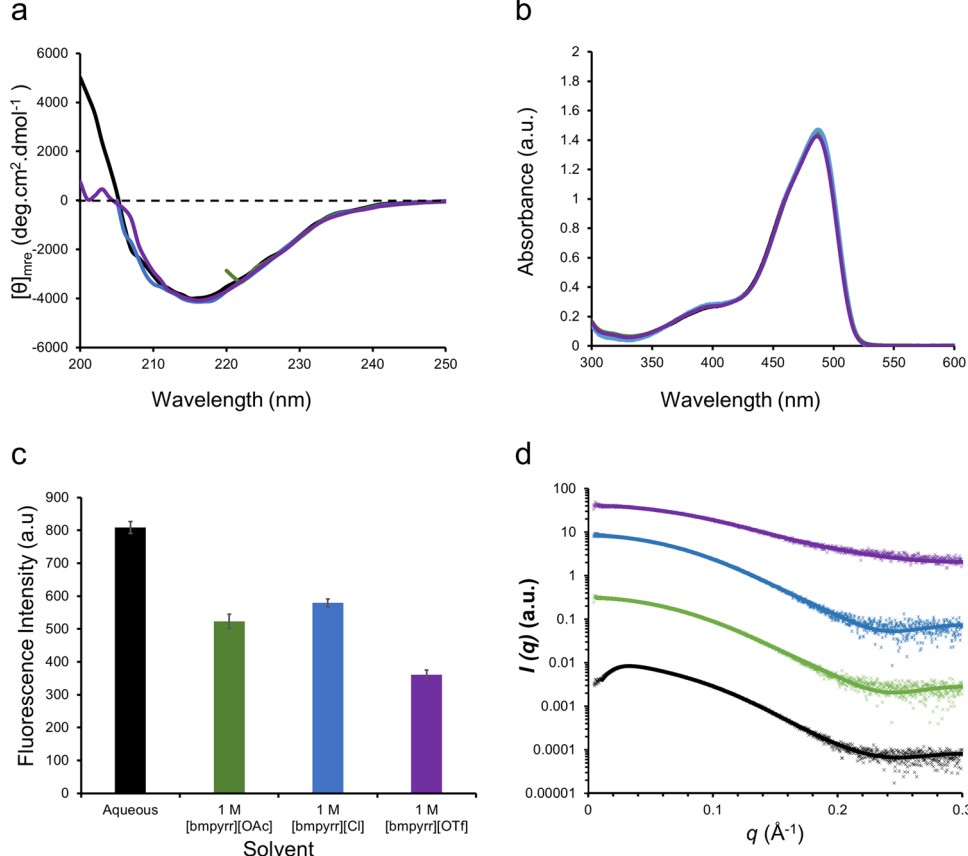

**Fig. 2 Secondary and tertiary structure of GFP in aqueous ionic liquid solutions. a** Far-UV CD spectra (mean residue ellipticity (mre) against wavelength) showing retention of $\beta$-sheet structure. **b** UV/Vis spectroscopy absorbance data (absorbance against wavelength) showing the GFP chromophore is in its native state. **c** Fluorescence intensity (a.u.) at 525 nm (error bars correspond to standard deviations, $n = 3$ — full data shown in Supplementary Fig. 1, and Supplementary Table 1), and (**d**) small-angle X-ray scattering (SAXS) profiles (separated for clarity) fitted with a cylinder model for GFP dissolved in water (black), 1 M [bmpyrr][OAc] (green), 1 M [bmpyrr]Cl (blue), and 1 M [bmpyrr][OTf] (purple) at 25 °C.

and CD measurements (Fig. 2c). In contrast to those results, fluorescence spectroscopy suggested that the ionic liquid was affecting GFP. In the presence of [bmpyrr] solutions, GFP fluorescence decreased with all anions, suggesting that the ionic liquids were causing a change in protein structure. The most significant decrease in fluorescence intensity (55%) occurred when [OTf]$^-$ was the anion, with lower decreases for the more polar anions, where Cl$^-$ and [OAc]$^-$ caused reductions of 28 and 38%, respectively. Similar results were observed with [bmim] solutions (Supplementary Fig. 2b). However, given the sensitivity of fluorescence to other environmental factors (such as solution pH), it could not be concluded that the observed changes were a result of any structural changes in GFP[26,27]. To probe this further, we performed small-angle X-ray scattering (SAXS) to investigate any potential changes to the global architecture of the protein (Fig. 2d). SAXS profiles for GFP were fitted to a cylindrical model, giving a diameter and length of 34.6 and 49.9 Å in water respectively, largely consistent with the crystal structure dimensions[28]. In [bmpyrr][OAc] and [bmpyrr]Cl solutions, there was a negligible change in the GFP dimensions (Supplementary Table 2). However, for GFP in the [bmpyrr][OTf] solution, both the diameter and length contracted to 26.3 and 44.3 Å, respectively. This trend was also observed for the equivalent [bmim] salts (Supplementary Fig. 2c, and Supplementary Table 2). The SAXS data was therefore in broad agreement with the fluorescence results, where [OTf]$^-$ salts had the most substantial effect on the tertiary structure of GFP. In this case, we measured a contraction in the GFP dimensions, which was likely to be the

cause of the change in fluorescence. The conclusions that the cation seemingly had no bearing on the interactions with the protein, with the anion effect being dominant, is consistent with previous studies[29–31].

**Surface mapping of ionic liquid–protein interactions**. It was therefore clear that spectroscopic and SAXS measurements alone failed to conclusively determine the specific molecular effects of the ionic liquids on protein structure. To investigate the effects of [OTf]$^-$ in greater detail, we turned to NMR spectroscopy. $^1$H-$^{15}$N heteronuclear single quantum coherence (HSQC) NMR spectroscopy was used to gain atomistic insight into the location of the ionic liquid interactions with GFP (Fig. 3a–c and Supplementary Fig. 10–13). HSQC plots of GFP in solutions of [bmim][OAc], [bmim]Cl, and [bmpyrr][OTf] were analysed and compared to that of aqueous GFP. From this analysis, up to 42 individual amino acid residues were observed to show distinct changes in chemical shifts in response to the presence of ionic liquid (Fig. 3). Inspection of the location of these residues within the crystal structure of GFP revealed that they were predominately located at either the ends of the $\beta$-barrel motif or within the central cavity of the protein (Fig. 3d–i). These locations represent the more flexible regions of the protein, and hence would be expected to be more susceptible to changes in the environment and to interact with dissolved ions. Comparing the chemistry of the residues that shifted (in terms of charged, polar, amphiphilic, or hydrophobic) to the primary sequence suggested

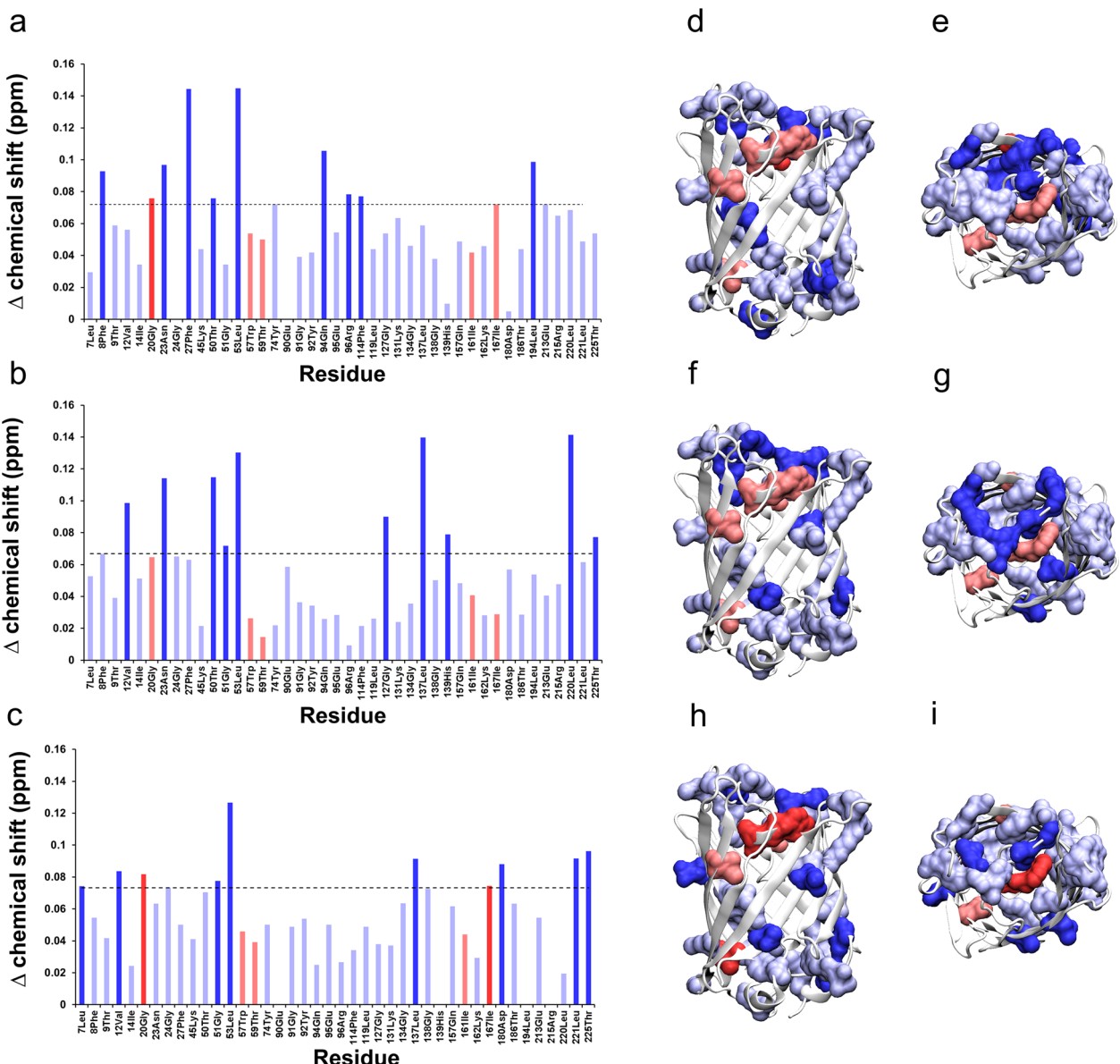

**Fig. 3 HSQC NMR surface mapping of ionic liquid–protein interactions. a–c** Change in chemical shift for GFP amino acid residues (as measured by $^1$H-$^{15}$N HSQC NMR spectroscopy — Supplementary Fig 8) comparing D$_2$O solutions to 1M solutions of [bmim][OAc] (**a**), [bmim]Cl (**b**), [bmpyrr][OTf] (**c**). **d–i** Corresponding models showing GFP from the side (**d**, **f**, **h**) and from the top (**e**, **g**, **i**) highlighting the shifting amino acids in the presence of [bmim][OAc] (**d**, **e**), [bmim]Cl (**f**, **g**), [bmpyrr][OTf] (**h**, **i**). Residues colored in blue are solvent accessible, and those colored in red are not. The darker colors represent the top ten residues that displayed the greatest shift in chemical shift. Models were created using the PDB ID: 1ema[41] with molecular graphics programme, VMD[42].

that they were broadly representative of the residue make-up of the whole protein. While this would initially suggest that there was no over-arching chemical trend in which amino acids the ionic liquids were interacting with, a closer examination of the magnitude of the change in chemical shift revealed subtle differences between the ionic liquids.

In the presence of [OAc]$^-$, the GFP residues that shifted the most were predominantly located on the interior of the protein with no bias towards polar or hydrophobic side chains (Fig. 3a, d, e). Similarly, in the presence of Cl$^-$, there was no bias towards polar or hydrophobic residues or any obvious preference for accessible or internal residues (Fig. 3b, f, g). In contrast, [OTf]$^-$ caused the greatest changes in chemical shifts in hydrophobic residues on the surface of the protein (Fig. 3c, h, i). This indicated that the more interacting anions ([OAc]$^-$ and Cl$^-$) might be

causing subtle changes in the protein that were unobservable through bulk structure characterisation. The least interacting anion tested, [OTf]$^-$, appeared to interact directly with the protein surface. This, in part, could at least explain why the structure was more obviously perturbed by the [OTf]$^-$ ionic liquids: structural contraction driven by explicit surface interactions.

HSQC-NMR allowed us to establish, alongside the spectroscopy and SAXS measurements, that the ionic liquids were indeed affecting the protein structure. However, it was still not clear whether this was due to a direct interaction with the ionic liquid moieties, or through some other mechanism. To delineate ionic liquid interactions with the protein, we used saturation transfer difference (STD)-NMR to investigate further (Fig. 4 and Supplementary Figs. 7–9). STD-NMR reveals site-specific

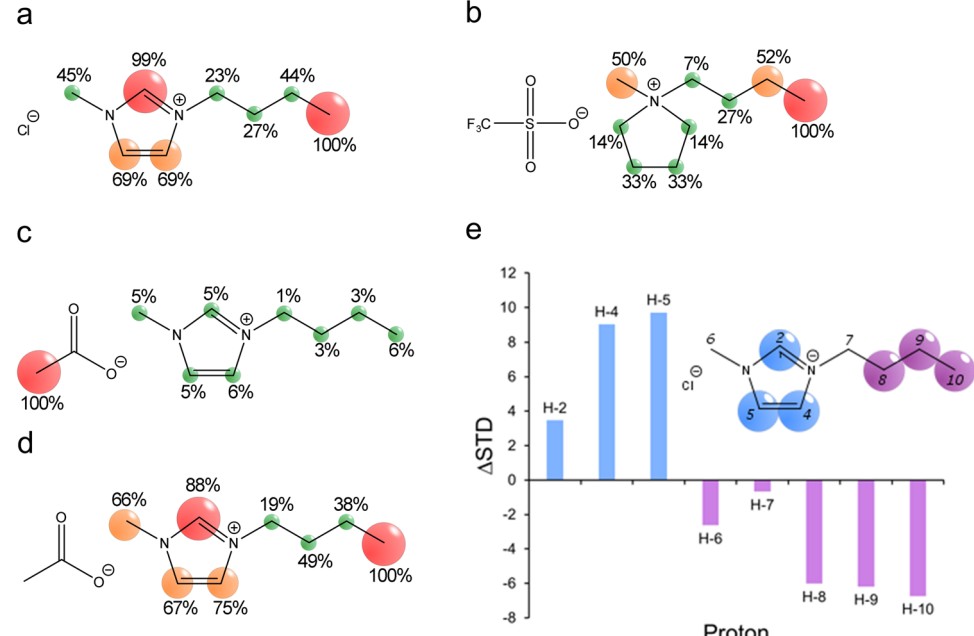

**Fig. 4 Determination of ion binding interactions with protein surface. a–d** STD-NMR Epitope maps of [bmim]Cl (**a**), [bmpyrr][OTf] (**b**) and [bmim][OAc] (**c,d**) relative to the most intense anion (**c**) and cation (**d**) signal. (**e**) DEEP-STD histogram (ΔSTD, 6.55/0.6 ppm) of [bmim]Cl and GFP at a ligand/protein ratio of 1000:1 in $D_2O$, with resulting differential epitope map of the numbered structure (aromatic interactions in blue, aliphatic interactions in purple).

interactions of ligand molecules by selectively saturating receptor macromolecules and measuring the saturation transfer to bound ligands via intramolecular $^1H–^1H$ cross-relaxation[32]. Here, the spatial proximity of ionic liquid protons with respect to GFP was determined, and interaction sites of [bmim][OAc], [bmim]Cl, and [bmpyrr][OTf] were visualised as epitope maps. The purpose was to determine that any structural changes were indeed a result of ionic liquid interactions.

**Ion binding interactions at the protein surface**. All of the ionic liquids produced STD signals, indicating direct interaction with GFP surface residues. When considering the signals with respect to the cation, the strongest interaction sites for [bmim] were the terminal –$CH_3$ group of the alkyl chain and the C-2-position of the [bmim] ring (Fig. 4a, d). There was very little difference in the STD signals of [bmim]Cl and [bmim][OAc]. This suggests that cation binding modality is independent of anion pairing. For the [bmpyrr] cation, in a response similar to that observed for [bmim], the terminal -$CH_3$ group gave the most intense STD response (Fig. 4b). Unlike [bmim] cations, protons around the charge centre of the pyrrolidinium ring gave comparatively lower STD responses, suggesting aromaticity plays a key role in ionic liquid–protein interactions. These differences suggested that [bmim] cations display a clear balance between electrostatic, hydrophobic, and aromatic interactions. However, [bmpyrr] cations display a subtler balance that appears to be somewhat in favor of hydrophobic interactions, rather than electrostatic interactions; this is likely a consequence of the more hydrophobic nature of the cation itself.

The -$CH_3$ group of the acetate anion provided an additional avenue of investigation for the binding of [bmim][OAc] to the protein surface. When considering the NMR signal for the anion as well as the cation, the strongest interaction site is the -$CH_3$ group of the acetate anion (Fig. 4c). This data was in broad agreement with the CD, UV/Vis (Fig. 2) and the HSQC NMR spectroscopy (Fig. 3), which all suggested that overall interactions between the ionic liquid and the protein surface were dominated

by the anion: a phenomenon frequently observed throughout the literature and confirmed here.

Having established the relative independence of the cation in binding to the protein surface, differential epitope mapping (DEEP)-STD between [bmim]Cl and GFP was carried out to determine whether there was any discrimination between what types of surface residues the hydrophobic and electrostatic groups of the cation interacted with (Fig. 4e). Irradiation of the complex with frequencies associated with aromatic (6.55 ppm) and aliphatic (0.6 ppm) residues (i.e., selective probing) revealed that the aromatic protons of the [bmim] ring interact with aromatic residues of the protein to different extents. The C-4 and C-5 protons gave positive STD shifts (ΔSTD) 3 times greater than the C-2 proton (Fig. 4e). This suggested that the back of the [bmim] ring interacted with aromatic residues more strongly than the front. Similarly, the end of the aliphatic carbon chain (i.e., the protons on C-8, C-9, and C-10) preferentially interacted with aliphatic residues, while the N-bound protons exhibited very little ΔSTD.

**Thermal stability of GFP in aqueous ionic liquids solutions**. Finally, having now established through a suite of static-structure probing techniques that protein structure is perturbed by ionic liquids, we sought to consolidate these observations to establish how these minor structural changes affected the protein stability. We therefore turned to temperature-dependent studies to determine how the presence of ionic liquids impacted the thermodynamics of protein stabilisation (Fig. 5 and Table 1).

Temperature-dependent CD spectroscopy (Fig. 5a, Supplementary Fig. 3) was used to determine thermal denaturation (with respect to secondary structure) for GFP in water and aqueous [bmpyrr] solutions (Fig. 5b). The secondary structure thermal stability in all cases was determined from the progressive reduction in intensity at 230 nm (Supplementary Fig. 3). GFP was most stable in water, with a half-denaturation temperature ($T_m$) of 87.0 °C (Table 1). In the presence of ionic liquid, GFP was less stable in all cases with half-denaturation temperatures of 84.5,

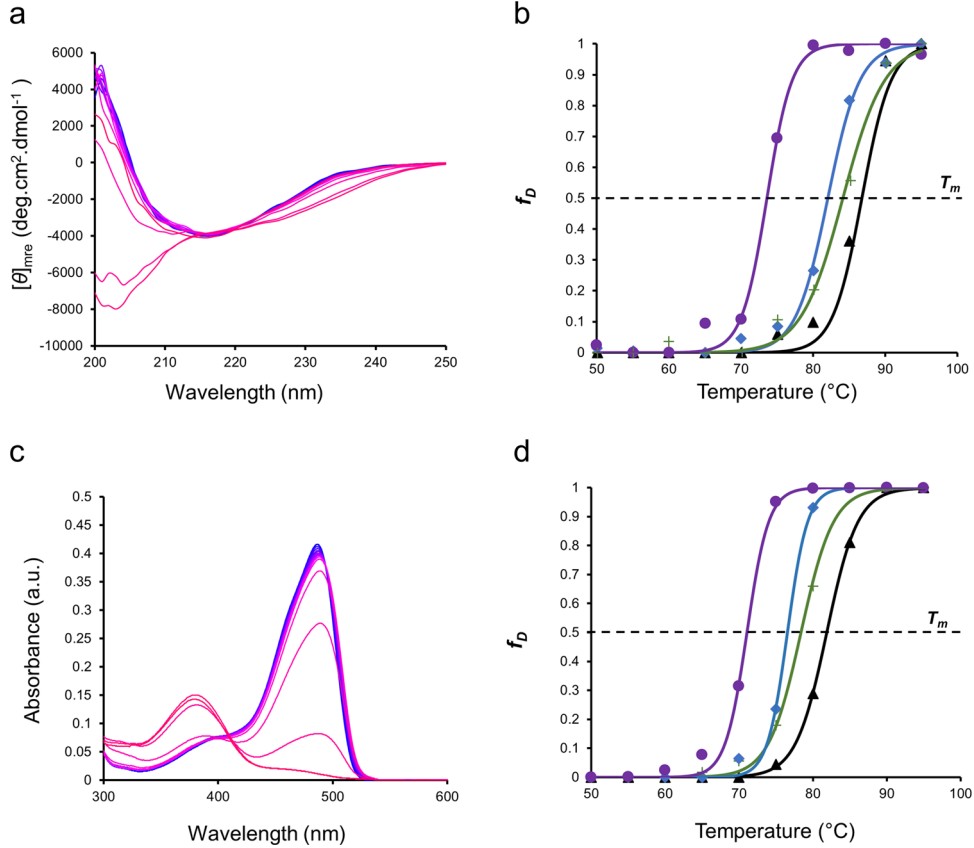

**Fig. 5 Thermal stability of GFP in aqueous ionic liquids solutions. a,c** Temperature-dependent CD (**a**) and UV/Vis (**c**) showing thermal denaturation of aqueous GFP. **b,d** Plots of fraction denatured as calculated using a two-state model of denaturation from CD spectroscopy [**b** data from Fig. 5a and Supplementary Fig. 3] and UV/Vis spectra [**d** data from Fig. 5c and Supplementary Fig. 4], for GFP in pure water (black triangles), 1 M [bmpyrr][OAc] (green pluses), 1 M [bmpyrr]Cl (blue diamonds), 1 M [bmpyrr][OTf] (purple circles). Solid lines represent data fitted with sigmoid.

**Table 1 Thermodynamic parameters for GFP denaturation in aqueous solutions.**

| Solution | $T_m$ (°C) | $\Delta H_m$ (kJ mol$^{-1}$) | $\Delta S_m$ (J K$^{-1}$ mol$^{-1}$) | $\Delta\Delta G_D$ (H$_2$O)$^a$ (kJ mol$^{-1}$) |
|---|---|---|---|---|
| CD | | | | |
| Aqueous | 87.0 ± 0.3 | 96.3 ± 9.9 | 1119 ± 303 | 0 |
| 1 M [bmpyrr][OAc] | 84.5 ± 0.4 | 102.0 ± 2.6 | 1219 ± 164 | −2.9 ± 0.4 |
| 1 M [bmpyrr]Cl | 81.2 ± 0.5 | 108 ± 10.4 | 1347 ± 352 | −8.1 ± 2.1 |
| 1 M [bmpyrr][OTf] | 73.3 ± 0.2 | 95.2 ± 0.2 | 1296 ± 50 | −16.3 ± 0.7 |
| UV/Vis | | | | |
| Aqueous | 82.0 ± 0.1 | 104.9 ± 0.3 | 1282 ± 57 | 0 |
| 1 M [bmpyrr][OAc] | 78.7 ± 0.2 | 76.8 ± 2.4 | 976 ± 146 | −3.0 ± 0.4 |
| 1 M [bmpyrr]Cl | 77.3 ± 0.1 | 92.3 ± 5.7 | 1201 ± 252 | −6.0 ± 1.2 |
| 1 M [bmpyrr][OTf] | 71.1 ± 0.2 | 109.4 ± 8.1 | 1561 ± 358 | −18.3 ± 3.7 |
| 1 M [bmim][OAc] | 75.6 ± 0.3 | 88.7 ± 6.2 | 1210 ± 270 | −10.2 ± 2.0 |
| 1 M [bmim]Cl | 73.6 ± 0.2 | 65.1 ± 4.5 | 883 ± 197 | −7.2 ± 1.4 |
| 1 M [bmim][OTf] | 70.1 ± 0.3 | 71.2 ± 0.01 | 1014 ± 10 | −11.7 ± 0.1 |

$^a$Free energy of denaturation calculated at the corresponding $T_m$ of GFP in water.

81.2, and 73.3 °C for [bmpyrr][OAc], [bmpyrr]Cl, and [bmpyrr][OTf] respectively (Fig. 5b and Table 1). Plots of free energy against temperature — as calculated from the temperature-dependent CD plots — were used to determine the thermodynamic parameters for denaturation (Supplementary Fig. 6 and Table 1). For GFP in [bmpyrr][OAc] and [bmpyrr]Cl the small reductions in $T_m$ were a result of slight increases in enthalpy of denaturation ($\Delta H_m$) being offset by increases in entropy of denaturation ($\Delta S_m$). Further analysis determined that this

reduction in stability, relative to GFP in water, equated to differences in free energy of denaturation ($\Delta\Delta G_D$) values of −2.9 and −8.1 kJ mol$^{-1}$, respectively. In the presence of [bmpyrr][OTf], the comparatively larger reduction in stability of GFP was due to a reduction in $\Delta H_m$ that was concomitant with an increase $\Delta S_m$, which accumulated in a destabilisation of −16.3 kJ mol$^{-1}$ (Table 1). These observations were broadly in line with our other experiments, specifically SAXS measurements detailing that [bmpyrr][OTf] caused a contraction of GFP (contributing to

the increase in $\Delta H_m$) and the HSQC NMR, which indicated that $[\mathrm{OTf}]^-$ preferentially interacted with hydrophobic residues (reducing the enthalpic barrier to denaturation).

Temperature-dependent UV/Vis spectroscopy was used to investigate the stability of GFP with respect to its tertiary structure. This was determined through monitoring the decrease in absorbance at 487 nm (Fig. 5c and Supplementary Fig. 4), as this absorbance is highly sensitive to the global conformation of the protein[33]. As expected, the thermal stability of GFP with respect to its tertiary structure — in all cases — was lower than the equivalent secondary structure (Fig. 5d and Table 1). Following the same trend as observed using CD, in aqueous solutions the $T_m$ was 82 °C, which reduced to 78.7, 77.3, and 71.1 °C in the presence of [bmpyrr][OAc], [bmpyrr]Cl, and [bmpyrr][OTf] respectively (Table 1). These corresponded to destabilisations ($\Delta\Delta G_D$) of −3, −6, and −18.3 kJ mol$^{-1}$, reflecting what was observed for the secondary structure.

Unlike the parameters calculated from CD measurements, monitoring of the tertiary structure denaturation revealed nuances in the destabilisation caused by the different ionic liquids (Table 1). Here, [bmpyrr][OAc] and [bmpyrr]Cl destabilised the protein predominately by reducing the enthalpic barrier to tertiary structure denaturation, whilst [bmpyrr][OTf] increased the entropic gain from denaturation. When [bmim] was used as the cation, similar results were obtained (Table 1 and Supplementary Fig. 5), ruling out any significant contribution from the cation. This was consistent with our other results, and previous studies, that ionic liquid–protein interactions are driven by the anion[29–31]. In terms of the thermal stability of GFP in response to ionic liquids, it was evident that the more interacting nature of $[\mathrm{OAc}]^-$ and $\mathrm{Cl}^-$ was causing more of a destabilisation of the interactions holding together the tertiary structure, than those of the highly stable β-barrel secondary structure motif. Conversely, $[\mathrm{OTf}]^-$ caused both a contraction in the protein size and reduced the barrier to unfolding, a direct result of preferential interaction with the protein surface, particularly the hydrophobic residues.

In conclusion, using GFP as an archetypal probe, we have shown that the interactions between proteins and ionic liquids are of significant complexity and require multiple complementary lines of enquiry to draw robust conclusions. Spectroscopic and SAXS measurements established that both the secondary and tertiary structure of the protein were affected by the presence of ionic liquids. Furthermore, STD-NMR experiments confirmed that structural changes were a direct consequence of the ionic liquids interacting with the protein surface. However, alone, these observations could not fully explain the true impact the ionic liquids had on the protein. This required investigating the thermal denaturation of GFP, which revealed the full impact of ionic liquids on the structure and stability of the protein.

We have therefore demonstrated the complexity of ionic liquid–protein interactions, and the implications on stability, showing that a multi-technique approach is ideal for gaining a full understanding, and that it is difficult to derive sound conclusions on ionic liquid–protein interactions through single pieces of evidence. This difficulty is in part due to similarity of ionic liquids properties, and structures, that necessitates a balance between in-depth and comprehensive investigations. Importantly, it is vital to know how the ionic liquid binds, where it binds, the resulting effect on protein structure, and how this affects protein stability. Improvements in ionic liquid design or protein mutation could then be targeted to rapidly advance non-aqueous biocatalysis.

Given the vast heterogeneity in the structures of proteins in terms of secondary structure motifs and amino acids present on the surface, it is expected that different proteins will behave differently to the plethora of ionic liquids available. This study shows that the anion plays a vital role in determining the nature of ionic liquid–protein interactions. Although the anion range is limited, we have shown that these complementary techniques provide an analytical framework for expanding on this study to fully delineate the role of ionic liquid structure on interactions with proteins. Frameworks and methodologies such as this are critical to a holistic understanding and should be applied to a larger range of proteins and ionic liquids for establishing the best strategies to advance non-aqueous biocatalysis.

## Methods

**GFP expression.** Wild-type superfolder green fluorescent protein (GFP, codon-optimised for Escherichia coli [E. coli] expression) with an N-terminal 6X poly-histidine tag was synthesised by GeneArt (ThermoFisher) and provided in an expression plasmid with a T7 promoter driving expression. The plasmid was transformed into E. coli Tuner (DE3) cells (Novagen) and single colonies were used to inoculate precultures in Lysogeny broth (LB) supplemented with 50 µg mL$^{-1}$ kanamycin. The precultures were grown overnight at 37 °C with shaking at 200 rpm. For fluorescence measurements, precultures were diluted 1:100 in 1 L of fresh LB with 50 µg mL$^{-1}$ kanamycin in a 2 L baffled flask and grown to an OD$_{600}$ of 0.6. Expression was induced by the addition of isopropyl β-D-1-thiogalactopyranoside (IPTG) to a final concentration of 500 µM. Cells were grown overnight at 20 °C.

Expression of $^{15}$N-labelled GFP for NMR studies followed a high cell density IPTG induction method using the minimal medium as described in Sivashanmugam, et al.[34]. Briefly, cells were grown in LB medium until an optical density at 600 nm (OD$_{600}$) of ~5 was achieved with cells harvested and resuspended in minimal medium (50 mM Na$_2$HPO$_4$, 25 mM KH$_2$PO$_4$, 10 mM NaCl, 5 mM MgSO$_4$, 0.2 mM CaCl$_2$, 0.25X trace metal solution, 0.25 × Basal Medium Eagle (BME) Vitamins, 0.1% $^{15}$NH$_4$Cl, 1.0% $^{13}$C-glucose, pH 8.2) to the same OD. Cells were grown for 1 h at 37 °C with shaking at 200 rpm, followed by a temperature downshift to 20 °C and induction with IPTG to a final concentration of 500 µM. Cells were harvested at OD$_{600}$ of 20 for protein purification.

Following expression, cells were harvested by centrifugation at $1500 \times g$ for 20 min and resuspended in 20 mL of lysis buffer (50 mM Tris-HCl pH 8.0, 20 mM imidazole and 2 M NaCl) supplemented with 1 mg mL$^{-1}$ lysozyme (Sigma-Aldrich), and incubated at room temperature for 1 h. Phenylmethylsulfonyl fluoride (PMSF, Sigma-Aldrich) was added to a final concentration of 1 mM. Cells were subjected to freeze fracture at −80 °C, and then further lysed by sonication for 10 min (20 s on, 20 s off, 50% amplitude). The cell lysates were clarified by centrifugation at $35,000 \times g$ for 30 min to remove cell debris and insoluble protein. Clarified lysates were purified using the ÄKTA Pure system (GE Healthcare) with a 5 mL HisTrap column (GE Healthcare) according to the manufacturer's instructions with an imidazole elution gradient using lysis buffer supplemented with 500 mM imidazole. Samples were dialysed against 50 mM sodium phosphate buffer to remove most of the salt and concentrated using a Vivaspin centrifugal concentrator with a 10 kDa molecular weight cutoff (MWCO) before storage at 4 °C.

**Nuclear magnetic resonance (NMR) spectroscopy experiments.** All Saturation-Transfer Difference (STD)-NMR experiments were conducted at 25 °C on a Bruker Avance III HD 600 MHz spectrometer equipped with a triple-resonance cryoprobe. All STD-NMR experiments were measured with ionic liquid:protein ratio of 1000:1 (ionic liquid concentration = 50 mM, GFP concentration = 50 µM) in D$_2$O. For protein saturation, a series of Eburp2.1000 shaped 90° pulses was used with a total protein saturation time of 1.5 s. The offset of the saturation pulse on the protein was set to −169.3 Hz (−0.28 ppm, on-resonance) and to 20,000 Hz (33.33 ppm, off-resonance), respectively. The spectra were acquired using a time domain (TD) of 32k (F2) and 64 scans in a spectral window of 16 ppm centered at 4.69 ppm (water signal). A Sinc1.1000 shaped 90° pulse (2 ms) was used for solvent sup-pression. A spinlock filter with a length of 20 ms was applied to suppress protein background signals. The STD effect was calculated by $(I_0 - I_{STD})/I_0$, in which $I_0$ is the peak intensity in the off-resonance spectrum and $(I_0 - I_{STD})$ is the peak intensity in the STD spectrum[35–37]. The STD intensity of the largest STD effect was set to 100%, and the relative intensities were calculated according to this.

For the Differential Epitope Mapping (DEEP)-STD experiments[38], two different on-resonance frequencies were used, i.e. 360.0 Hz (0.60 ppm) for irradiation of the aliphatic region, and 3930.0 Hz (6.55 ppm) for irradiation of the aromatic region. The DEEP-STD factor ($\Delta\mathrm{STD}_i$) is then calculated according to the following equation:

$$\Delta\mathrm{STD}_i = \frac{\mathrm{STD}_{\mathrm{exp1},i}}{\mathrm{STD}_{\mathrm{exp2},i}} - \frac{1}{n}\sum_{i}^{n}\left(\frac{\mathrm{STD}_{\mathrm{exp1},i}}{\mathrm{STD}_{\mathrm{exp2},i}}\right) \tag{1}$$

In this equation, $\mathrm{STD}_{\mathrm{exp1},i}$ is the STD amplification factor in the experiment with irradiation of the aromatic region, and $\mathrm{STD}_{\mathrm{exp2},i}$ is the STD amplification factor in the experiment with irradiation of the aliphatic region.

$^1$H–$^{15}$N Heteronuclear Single Quantum Coherence (HSQC) spectra were recorded on a Bruker Avance III HD 800 MHz equipped with 5 mm TCI

cryoprobe. All spectra were recorded with 1536 complex $t_1$ increments of 100 complex data points and 8 transients. The spectral widths were 36 ppm and 20 ppm for the $^{15}N$ ($F_1$) and $^1H$ ($F_2$) dimensions. The sample concentration is 50 µM GFP in 1 M ionic liquid solutions with 90/10% (v/v) $H_2O/D_2O$ at 310 K. The data were processed using Bruker TopSpin software (https://www.bruker.com/).

**Other spectroscopy experiments**. UV/Vis spectroscopy was performed on a Shimadzu UV2600 fitted with a Peltier temperature controller. Aqueous solutions of GFP (0.2–1.1 mg mL$^{-1}$) and ionic liquid were measured in quartz cuvettes sealed with PTFE stoppers with a pathlength of 10 mm. Samples were incubated for 120 s for temperature intervals of 5 °C between 25 and 95 °C with a tolerance of 0.2 °C. The protein concentration was determined using an extinction coefficient of $\varepsilon_{280\,nm} = 0.685$ mg mL$^{-1}$ cm$^{-1}$.

Circular dichroism experiments were performed on an Applied Photosystem Chirascan Spectropolarimeter with quartz cells ($l = 0.01$ cm) equipped with a Melcor MTCA temperature controller. Thermal denaturation curves were obtained from aqueous solutions, and aqueous ionic liquid mixtures (0.2–0.6 mg mL$^{-1}$) with a heating rate was set to 100 °C min$^{-1}$ and incubation of 120 s for temperature intervals of 5 °C between 25 and 95 °C with a tolerance of 0.2 °C. Spectra data were collected with 1 nm steps between 260 and 180 nm and 2 s collection time per step.

Fluorimetry experiments were completed on SpectraMax Gemini EM Microplate Spectrofluorometer (Molecular Devices, Sunnyvale, CA) using a 96-well microplate. GFP protein samples in aqueous samples (0.2 mg mL$^{-1}$) were pipetted (150 µL) into each well with at least three sample replicates for all measurements. The 96-well microplate was well mixed for 25 s before measuring the fluorescence of the samples. Samples were excited at a wavelength of 395 nm with a cutoff wavelength of 515 nm, and the emission spectra between 500 and 625 nm in steps of 5 nm at room temperature. Absolute fluorescence values were exported to Excel for subsequent data processing.

**X-ray scattering**. Small-angle X-ray Scattering (SAXS) measurements were performed at the Diamond Light Source (beamline B21) with an X-ray energy of 12.4 keV at a fixed camera length of 3.9 m. GFP protein samples in aqueous solutions (5 mg mL$^{-1}$) were loaded using the automated BIOSAXS robot at 15 °C and spectra was collected. B21 was operated to collect data between 0.015 and 0.3 Å$^{-1}$ on a 2 M Pilatus detector (Dectris, UK). A minimum of 28 spectra was collected for each sample and processed using the ScÅtter (http://www.bioisis.net/) for Guinier analysis and SasView (https://www.sasview.org/) to fit samples to a cylinder geometry.

**Thermodynamic calculations**. Thermodynamic parameters were calculated as described previously by Brogan, et al.[39]. Plots of $\Delta G_D$ against temperature (Supplementary Fig. 6) were used to calculate the $T_m$, $\Delta H_m$ and $\Delta S_m$ for GFP in the different aqueous environments. Using these same plots, by calculating $T\Delta S$ and $\Delta H$ at the $T_m$ for GFP in aqueous solution, it was possible to calculate the free energy ($\Delta\Delta G_D$) associated with destabilisation for GFP in response to the ionic liquids.

**Synthesis of ionic liquids**. [Bmpyrr] ionic liquids were synthesised and fully characterised by NMR spectroscopy and MS as reported previously[6]. [Bmim]Cl was synthesised according to existing literature procedures[40]. [Bmim][OTf] and [bmim] [OAc] ionic liquids were purchased from Sigma-Aldrich with a minimum purity of 97 and 96% respectively and used without further purification. Purity of [bmim] ionic liquids were assessed by $^1H$ NMR before use (Supplementary Figs. 7–9).

## Data availability
The authors declare that all data supporting the findings of this study are available within the article and Supplementary Information files, and from the corresponding author on request.

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

## Acknowledgements

The authors thank the EPSRC (Frontier Engineering Grant EP/K038648/1) and Future Vaccine Manufacturing Hub (EP/R013764/1) for financial support and John Televantos for sponsoring Liem Bui-Le's PhD. The authors would also like to thank Nikul Kunti for access to beamline B21 (main-in service) and Giuliano Siligardi for access to beamline B23 (Chirascan) at Diamond Light Source Ltd (Oxford, UK). Also, the authors would like to thank Dr Marc Morgan (CD spectroscopy) and Dr Yingqi Xu (NMR) at Imperial College London for access and help with measurements.

## Author contributions

J.P.H., K.M.P., A.B., L.B.-L., and A.P.S.B. designed the experiments. L.B.-L., C.J.C., J.A.J.A., and A.B. performed the experiments, and L.B.-L., C.J.C., A.B., and A.P.S.B analysed and discussed the data. The manuscript was written by L.B.-L., C.J.C, and A.P.S.B. All authors proof-read, discussed and have given approval to the final version of the manuscript.

## Competing interests

The authors declare no competing interests.
