## [Peer Review File · Communications Chemistry]

Reviewers' comments:

Reviewer #1 (Remarks to the Author):

The manuscript reports an experimental study of the interactions between selected ionic liquids and a model protein. Solutions of biomolecules in ionic liquids are a challenging and emerging research object as understanding their structure-property relationships can lead to significant advances in the fields of biocatalysis and separation science. In the present work, the IL-protein interactions are studied by a combination of analytical methods: circular dichroism, fluorescence, UV/vis and NMR spectroscopies as well as small angle x-ray scattering. These methods are well chosen as they allow insights into the interactions from different points of view. What came as a surprise though, was that no means of vibrational spectroscopy were applied. IR, Raman, VCD, and ROA spectroscopies are most frequently used to study molecular interactions and can deliver highly specific insights (see e.g. RSC ADVANCES Volume: 2 Issue: 32 Pages: 12329-12336 Published: 2012). Is there a reason, why vibrational spectroscopy was not considered? The manuscript should at least mention and explain this point.

Another point to be clarified is the impurities. How did the authors ensure that the UV/vis and fluorescence spectra are not affected or even governed by the impurities?

If these points can be clarified, the manuscript is suitable for publication in Communications Chemistry.

Reviewer #2 (Remarks to the Author):

The work entitled "Revealing the Complexity 1 of Ionic Liquid-Protein Interactions through a Multi-Technique Investigation" describes an interesting results. I would like to recommend the manuscript for publication with minor revision.

The main goal of this studies is to determine/describe the interaction between ionic liquids and proteins, based on model protein (GFP) and model ionic liquids [bmpyrr][OAc], [bmpyrr]Cl, and [bmpyrr][OTf].

The method to achieve the goal is novel and innovative. Analytical framework for a comprehensive investigation into the interactions between ionic liquids and proteins was provided using complex analytical tools (multi-technique approach): UV/Vis, fluorescence, circular dichroism, and NMR spectroscopies alongside small-angle X-ray scattering and thermal denaturation studies. The results can be of interest to others in the community and the wider field. The work is based on detailed experiments which can be reproduced.

This work comprehensively introduce the reader in the presented problem and describes with details all the necessary data. The properties of proteins, enzymes and ionic liquids are complex, and e.g. there is no universal method for predicting the appropriate ionic liquid for a specific bioprocess. Anyway, it would be beneficial for the readers if Authors could also discussed some problems concerning the fact that ionic liquids with very similar structure can exhibit different behaviour towards various enzymes and the same enzyme immobilized in the particular ionic liquid may have different activity in various reactions (e.g. New J. Chem., 2015, 39, 1315; Int. J. Biol. Macromol., 2014, 63, 244; Green Chem., 2002, 4, 147).

In the Results and Discussion part no information (arguments, discussion) concerning the method of selection of ionic liquids for this studies can be found. Why these three ionic liquids were chosen for this purpose? Authors claim, that it is known from literature that the cation seemingly has no bearing on the interactions with the protein, with the anion effect being dominant. Why [OAc]-, [OTf]- Cl- were chosen?

The main conclusions have been drawn. However the question how universal the conclusions are to truly determine the impact of ionic liquids on protein structures. If we change the structure of ionic liquid (mainly anion) the interplay between ionic liquids and proteins will be completely different? Please add some comments. The reviewer understand that it is difficult to make general conclusion but it would be worth to underline that this is the new complex method which can be used for this purpose.

Reviewer #3 (Remarks to the Author):

The manuscript submitted by Hallett and co-workers provides relevant scientific findings, namely by proving a comprehensive understanding of the interactions occurring between proteins and ionic liquids by combining several techniques (CD, fluorescence spectroscopy, UV/Vis, NMR, and SAXS). I do recommend the publication of this manuscript in Communications Chemistry after major revisions, as listed below:

- Authors should explain why GFP was used as the model protein. This is particularly relevant since authors state in the beginning of the abstract the importance of ionic liquids in biocatalysis. Accordingly, why a model enzyme was not used instead? Furthermore, by using an enzyme, additional results on the enzymatic activity could be given.
- I do recommend the authors to carry out additional studies with a couple of other relevant proteins/enzymes to give support to their findings. Although the revealing of the complexity of Ionic-Liquid-Protein Interactions is not compromised, authors claim that they aim to provide a comprehensive understanding of the interactions occurring between proteins and ionic liquids, which is however limited if only one protein is addressed in their study.
- Authors clearly explain in the abstract the pitfalls in the field and what they aim to overcome, but no indications on the main results, i.e. which specific interactions occur and those that dominate, are provided in the abstract. The type of ionic liquids investigated is not provided in the abstract as well.
- Why the authors did choose imidazolium- and pyrrolidinium-based ionic liquids, particularly in a time where cholinium-based ionic liquids are being largely investigated in the proteins/enzymes field?
- Why CD spectra were not acquired for the imidazolium-based ionic liquids? I do suppose that it is because of the absorption of the aromatic ring, but authors should provide this information in the beginning of their results. On the other hand, authors can induce the precipitation of the protein and carry out the CD assays after its resuspension in appropriate buffer solutions. For comparative purposes, authors also can do the precipitation of GFP for the pyrrolidinium-based ionic liquids. This approach would give, at least, some qualitative information and impact of all ionic liquids investigated in the protein structure by CD assays.

Authors' responses reviewers comments

Reviewer #1 (Remarks to the Author):

The manuscript reports an experimental study of the interactions between selected ionic liquids and a model protein. Solutions of biomolecules in ionic liquids are a challenging and emerging research object as understanding their structure-property relationships can lead to significant advances in the fields of biocatalysis and separation science. In the present work, the IL-protein interactions are studied by a combination of analytical methods: circular dichroism, fluorescence, UV/vis and NMR spectroscopies as well as small angle x-ray scattering. These methods are well chosen as they allow insights into the interactions from different points of view.

If these points can be clarified, the manuscript is suitable for publication in Communications Chemistry.

We thank the reviewer for taking the time to provide detailed feedback on our manuscript. We hope the responses detailed below will clarify the results and will satisfy the concerns that were raised.

Detailed comments

What came as a surprize though, was that no means of vibrational spectroscopy were applied. IR, Raman, VCD, and ROA spectroscopies are most frequently used to study molecular interactions and can deliver highly specific insights (see e.g. RSC ADVANCES Volume: 2 Issue: 32 Pages: 12329-12336 Published: 2012). Is there a reason, why vibrational spectroscopy was not considered? The manuscript should at least mention and explain this point.

We agree with the reviewer that vibrational spectroscopy can provide insights into structural properties of proteins, and we have previously measured FTIR spectra of enzymes in the absence of solvent (*J. Am. Chem. Soc.* 2016, *Nat. Chem.* 2018). However, in those papers, whilst we were able to confirm protein structure through presence of Amide I and II peaks, the full band was difficult to resolve due to interference from the ionic liquids, making full protein structure determination difficult. Therefore, for this paper, we decided to use NMR, as the level of structural data gained is in much greater detail e.g. the specific amino acids. This coupled with our other spectroscopic measurements meant that the use of vibrational spectroscopy was surplus to our requirements.

For clarification, the sentence that read:

“Using UV/Vis, fluorescence, circular dichroism, and NMR spectroscopies alongside small-angle X-ray scattering and thermal denaturation studies, we have thoroughly

investigated the specific and non-specific interactions between GFP and a range of 1 butyl 1 methylpyrrolidinium ([bmpyrr]) and 1 butyl 3 methylimidazolium ([bmim]) salts (Fig. 1b).”

Now reads:

“Using UV/Vis, fluorescence, circular dichroism, and NMR spectroscopies (for detailed structural analysis) alongside small-angle X-ray scattering and thermal denaturation studies, we have thoroughly investigated the specific and non-specific interactions between GFP and a range of 1 butyl 1 methylpyrrolidinium ([bmpyrr]) and 1 butyl 3 methylimidazolium ([bmim]) salts (Fig. 1b).”

Another point to be clarified is the impurities. How did the authors ensure that the UV/vis and fluorescence spectra are not affected or even governed by the impurities?

The bmpyrr ionic liquids were synthesised by previously established methods, with high purity validated by ^1H and ^{13}C NMR, and Mass Spectrometry in Brogan et al., Nature Chem, 2018. For the bmim ionic liquids, these were determined to be pure from ^1H NMR. Purity of the ionic liquids was already mentioned in the methodology section:

“Purity of [bmim] ionic liquids were assessed by ^1H NMR before use (Supplementary Fig. 7-9).”

As such, we feel that this point has already been adequately addressed.

Reviewer #2 (Remarks to the Author):

The work entitled “Revealing the Complexity 1 of Ionic Liquid-Protein Interactions through a Multi- Technique Investigation” describes an interesting results. I would like to recommend the manuscript for publication with minor revision.

The main goal of this studies is to determine/describe the interaction between ionic liquids and proteins, based on model protein (GFP) and model ionic liquids [bmpyrr][OAc], [bmpyrr]Cl, and [bmpyrr][OTf]. The method to achieve the goal is novel and innovative. Analytical framework for a comprehensive investigation into the interactions between ionic liquids and proteins was provided using complex analytical tools (multi-technique approach): UV/Vis, fluorescence, circular dichroism, and NMR spectroscopies alongside small-angle X-ray scattering and thermal denaturation studies. The results can be of interest to others in the community and the wider field. The work is based on detailed experiments which can be reproduced.

This work comprehensively introduce the reader in the presented problem and describes with details all the necessary data. The properties of proteins, enzymes

and ionic liquids are complex, and e.g. there is no universal method for predicting the appropriate ionic liquid for a specific bioprocess.

We thank the reviewer for taking the time to provide feedback on our paper, and for suggesting it is worthy of publication in Communications Chemistry. We hope the responses detailed below will adequately address the reviewer's points.

Detailed comments

Anyway, it would be beneficial for the readers if Authors could also discussed some problems concerning the fact that ionic liquids with very similar structure can exhibit different behaviour towards various enzymes and the same enzyme immobilized in the particular ionic liquid may have different activity in various reactions (e.g. New J. Chem., 2015, 39, 1315; Int. J. Biol. Macromol., 2014, 63, 244; Green Chem., 2002, 4, 147).

The perceived differences in behaviour of different enzymes with respect to ionic liquids was one of the key drivers behind why we wanted to pursue this study. To clarify this point further, we have added the following sentence with the Green Chem., 2002 reference as reference 15:

“This contradictory nature of how ionic liquids interact with proteins is manifested acutely in the activity of enzymes, where different behaviours may be observed depending on whether the enzyme is free, crosslinked, or immobilized on a solid support¹⁵.”

In the Results and Discussion part no information (arguments, discussion) concerning the method of selection of ionic liquids for this studies can be found. Why these three ionic liquids were chosen for this purpose? Authors claim, that it is known from literature that the cation seemingly has no bearing on the interactions with the protein, with the anion effect being dominant. Why [OAc]⁻, [OTf]⁻ Cl⁻ were chosen?

Previously literature and review papers such as Annu. Rev. Phys. Chem. 61, 63-83 (2010); Phys. Chem. Chem. Phys. 14, 415-426 (2012); and J. Am. Chem. Soc. 135, 5062-5067 (2013) (Ref. 26 – 28) state that anions are the dominant effect when determining protein and ionic liquid interactions. The three anions were picked as a broad range of water-soluble anions, which excludes anions such as [PF₆]⁻ and [NTf₂]⁻, and Kamlet-Taft β parameters of the [OAc]⁻, Cl⁻ and [OTf]⁻ salts demonstrate the selection of strongly coordinating, coordinating and non-coordinating anions, respectively.

A sentence has been added to the Results and Discussion to clarify the choice of anions with references Phys. Chem. Chem. Phys. 13, 16831-16840 (2011) and Chem. Lett. 38, 2-7, (2009) as reference 24 and 25 respectively:

“The anions selected for this study ([OAc]⁻, Cl⁻, and [OTf]⁻) were due to their water-soluble nature and the broad range of coordination ability as determined from Kamlet-Taft β parameters being 1.18, 0.87 and 0.49 for [OAc]⁻, Cl⁻, and [OTf]⁻ respectively²⁴⁻²⁵.”

The main conclusions have been drawn. However the question how universal the conclusions are to truly determine the impact of ionic liquids on protein structures. If we change the structure of ionic liquid (mainly anion) the interplay between ionic liquids and proteins will be completely different? Please add some comments. The reviewer understand that it is difficult to make general conclusion but it would be worth to underline that this is the new complex method which can be used for this purpose

We thank the reviewer for their suggestion and understand that there needs to be clarification on the limitations of the analytical framework in the conclusion.

We have expanded the conclusion with the following to highlight this point. The section that read:

“Given the vast heterogeneity in the structures of proteins in terms of secondary structure motifs and amino acids presented on the surface, it is to be expected that different proteins will behave differently to the plethora of ionic liquids available. These complementary techniques provide an analytical framework for future solvent-protein studies, ensuring a systematic deconstruction of interactions and effects. Frameworks and methodologies such as this are critical to a holistic understanding and necessary for establishing the best strategies to advance non-aqueous biocatalysis. “

Now reads:

“Given the vast heterogeneity in the structures of proteins in terms of secondary structure motifs and amino acids present on the surface, it is expected that different proteins will behave differently to the plethora of ionic liquids available. This study shows that the anion plays a vital role in determining the nature of ionic liquid-protein interactions. Although the anion range is limited, we have shown that these complementary techniques provide an analytical framework for expanding on this study to fully delineate the role of ionic liquid structure on interactions with proteins. Frameworks and methodologies such as this are critical to a holistic understanding and should be applied to a larger range of proteins and ionic liquids for establishing the best strategies to advance non-aqueous biocatalysis.”

Reviewer #3 (Remarks to the Author):

The manuscript submitted by Hallett and co-workers provides relevant scientific findings, namely by proving a comprehensive understanding of the interactions occurring between proteins and ionic liquids by combining several techniques (CD, fluorescence spectroscopy, UV/Vis, NMR, and SAXS). I do recommend the

publication of this manuscript in Communications Chemistry after major revisions, as listed below:

We thank the reviewer for their feedback on our paper, and we hope the responses detailed below will adequately address the reviewer's points.

Detailed comments

Authors should explain why GFP was used as the model protein. This is particularly relevant since authors state in the beginning of the abstract the importance of ionic liquids in biocatalysis. Accordingly, why a model enzyme was not used instead? Furthermore, by using an enzyme, additional results on the enzymatic activity could be given.

We wish to thank the reviewer for their suggestion; we selected GFP as it is one of the most well understood and fully characterised proteins in the literature to-date. Therefore, this provided us with a robust platform for investigating the effect of ionic liquids on protein structure through resolving specific amino acids signals in sophisticated techniques such as HSQC NMR, which would not be possible in lesser studied proteins. Furthermore, the intrinsic fluorophore of GFP allows for analysis of the tertiary structure through UV-vis and fluorescence spectroscopy. To clarify this we have now expanded the introduction from:

“Here, using green fluorescent protein (GFP) as an illustrative example, we provide an analytical framework for a comprehensive investigation into the interactions between ionic liquids and proteins (Fig. 1a).”

To

“Here, taking full advantage of the highly characterized – in terms of secondary and tertiary structure – green fluorescent protein (GFP), we provide an analytical framework for a comprehensive investigation into the interactions between ionic liquids and proteins (Fig. 1a).”

Similarly, we opted to not study an enzyme as structural analysis of enzymes are typically less robust than what is available for GFP. Although we do agree with the reviewer that measuring activity would be a great addition to the article, we believe it to be beyond the scope of this article. When considering enzyme activity, it will be necessary to decouple the effect of the ionic liquid components on the substrate, intermediate, and product (in terms of solvation, polarity, stability) from the impact on the protein. In light of this complexity, we decided to develop the robust analytical framework first (this paper), before moving towards investigating the impact of ionic liquids on enzyme activity.

I do recommend the authors to carry out additional studies with a couple of other relevant proteins/enzymes to give support to their findings. Although the revealing of

the complexity of Ionic-Liquid-Protein Interactions is not compromised, authors claim that they aim to provide a comprehensive understanding of the interactions occurring between proteins and ionic liquids, which is however limited if only one protein is addressed in their study.

Whilst we agree with the reviewer that additional studies would give additional support to the work presented here, the work required to do so is far beyond what is feasible for a communication, and would be considered for follow on studies (see above comment). This study required; the expression of the protein using ¹⁵N-labelled reagents, high-resolution (800 and 600 MHz) 2D and STD NMR time, and successful synchrotron applications for SAXS experiments. As such, we do not believe adding other examples is a feasible request for a single paper. Starting with a highly characterised protein such as GFP provides an excellent benchmark for establishing generalized rules as a body of future work, including our own, which could be then focussed on proteins or enzymes for specific applications, consistent with the general framework we have developed here. To clarify this, we have now stated in the article that

“we provide an analytical framework for a comprehensive investigation into the interactions between ionic liquids and proteins”

Authors clearly explain in the abstract the pitfalls in the field and what they aim to overcome, but no indications on the main results, i.e. which specific interactions occur and those that dominate, are provided in the abstract. The type of ionic liquids investigated is not provided in the abstract as well.

We thank the reviewer for the suggestion; we have altered the abstract to detail the type of anions and ionic liquids studied as well as an indication of main results.

The abstract has been changed from:

“Ionic liquids offer exciting possibilities for biocatalysis as solvent properties provide rare opportunities for customizable, energy-efficient bioprocessing. Unfortunately, proteins and enzymes are generally unstable in ionic liquids and several attempts have been made to explain why; however, a comprehensive understanding of the interactions between proteins and ionic liquids remains elusive. Here, we present an analytical framework (CD, fluorescence spectroscopy, UV/Vis, NMR, and SAXS) to probe the interactions, structure, and stability of a model protein (GFP) in ionic liquids. We demonstrate that protein stability requires a holistic perspective driven by similar analytical frameworks, as opposed to single-technique assessments that can deliver misleading conclusions. We also reveal unprecedented information regarding site-specific ionic-liquid protein interactions from the point of view of the strongest interactions on both the protein and the ionic liquid. Robust frameworks and methodologies such as this are critical to advancing non-aqueous biocatalysis and avoiding pitfalls associated with single-technique investigations.”

To

“Ionic liquids offer exciting possibilities for biocatalysis as solvent properties provide rare opportunities for customizable, energy-efficient bioprocessing. Unfortunately, proteins and enzymes are generally unstable in ionic liquids and several attempts have been made to explain why; however, a comprehensive understanding of the ionic liquid-protein interactions remains elusive. Here, we present an analytical framework (CD, fluorescence spectroscopy, UV/Vis, NMR, and SAXS) to probe the interactions, structure, and stability of a model protein (GFP) in a range (acetate, chloride, triflate) of pyrrolidinium and imidazolium salts. We demonstrate that protein stability requires a similar holistic analytical framework, as opposed to single-technique assessments that provide misleading conclusions. We reveal unprecedented information on site-specific ionic liquid-protein interactions, revealing that triflate (the least interacting anion) induces a contraction in the protein size that reduces the barrier to unfolding. Robust frameworks such as this are critical to advancing non-aqueous biocatalysis and avoiding pitfalls associated with single-technique investigations.”

Why the authors did chose imidazolium- and pyrrolidinium-based ionic liquids, particularly in a time where cholinium-based ionic liquids are being largely investigated in the proteins/enzymes field?

Cholinium-based ionic liquids are used in the protein/enzyme field due to their high biocompatibility, but we wanted to develop an analytic framework with well established systems, such as imidazolium and pyrrolidinium ionic liquids. Cholinium-amino acid ionic liquids were something that was considered by our group; however, decoupling the cholinium amino acids from the CD spectra remains challenging due to high CD signal from the amino acid anions. Regardless, future work in our group will be considering these ionic liquids alongside deep eutectic solvents as biocompatible alternatives to the ones we have chosen here.

Why CD spectra were not acquired for the imidazolium-based ionic liquids? I do suppose that it is because of the absorption of the aromatic ring, but authors should provide this information in the beginning of their results. On the other hand, authors can induce the precipitation of the protein and carry out the CD assays after its resuspension in appropriate buffer solutions. For comparative purposes, authors also can do the precipitation of GFP for the pyrrolidinium-based ionic liquids. This approach would give, at least, some qualitative information and impact of all ionic liquids investigated in the protein structure by CD assays.

We thank the reviewer for highlighting that there needs to be some clarification regarding why CD spectra was not been acquired for imidazolium-based ionic liquids. The reason is exactly as the reviewer suggested, which is due to the high absorption of the imidazolium moiety. To clarify this in the manuscript, we have added in a sentence in the Result and Discussion that reads:

“For CD spectroscopy we were limited to pyrrolidinium salts, as imidazole has significant absorbance in the far-UV region.”

In addition, for the precipitation experiments, the precipitation effect on structure vs. ionic liquid interactions would need to be decoupled. The room temperature CD and UV-vis spectra of GFP in water, buffer and aqueous ionic liquid solutions showed that there was no significant difference. In addition, for the temperature-dependent UV and CD spectra, a two-state denaturation was confirmed through the presence of isosbestic and isodichroic points. As such, ionic liquid interactions were assumed to be reversible and removing the ionic liquid and resuspension in buffer solutions would show the same spectra compared to if the protein was dissolved in buffer solution first.

REVIEWERS' COMMENTS:

Reviewer #1 (Remarks to the Author):

The authors have addressed the points raised and improved the manuscript significantly. It is now suitable for publication.

Reviewer #2 (Remarks to the Author):

All my suggestions and questions have been satisfactorily addressed and now this work in my opinion is suitable for publication.

Reviewer #3 (Remarks to the Author):

Authors properly considered the reviewers' comments and improved the manuscript accordingly. I do suggest the manuscript publication in its current form.